# Association between life events and later depression in the population-based Heinz Nixdorf Recall study—The role of sex and optimism

**Janine Gronewold**[1]*, **Ela-Emsal Duman**[1], **Miriam Engel**[2], **Miriam Engels**[3],
**Johannes Siegrist**[3], **Raimund Erbel**[2], **K-H. Jöckel**[2], **Dirk M. Hermann**[1]*

**1** Department of Neurology and Center for Translational Neuro- and Behavioral Sciences (C-TNBS), University Hospital Essen, University Duisburg-Essen, Essen, Germany, **2** Institute of Medical Informatics, Biometry and Epidemiology, University of Duisburg-Essen, Essen, Germany, **3** Institute of Medical Sociology, Medical Faculty, Heinrich-Heine-University Düsseldorf, Düsseldorf, Germany

* janine.gronewold@uk-essen.de (JG); dirk.hermann@uk-essen.de (DMH)

## Abstract

### Background

The association between life event stress and depressive symptoms has not been analyzed in the general population before.

### Methods

In the population-based Heinz Nixdorf Recall study, we assessed the association of 1.) the presence of important life events and 2.) life event stress, with the amount of depressive symptoms in univariable linear regressions and in multivariable regressions adjusted for age and sex (model 1) and age, sex and optimism as important determinants of coping with life events (model 2). Presence of life events and life event stress were assessed with the Social Readjustment Rating Scale (SRRS), optimism with the Life Orientation Test-Revised (LOT-R), and depressive symptoms with the 15-item Center for Epidemiological Studies Depression Scale (CES-D).

### Results

Of the total cohort of 4,814 participants, 1,120 had experienced important life events during the previous 6 months. Presence of important life events was significantly associated with higher CES-D scores (B = 2.6, 95%CI = 2.2 to 3.0, p < .001; model 2) compared to absence of life events. Associations were stronger for women than for men and for pessimists than for optimists. Among the participants with important life events, median (Q1; Q3) stress-score was 45.0 (39.0; 63.0). Stress-scores >Q3 were significantly associated with higher CES-D scores (2.2, 1.1 to 3.3, < .001) with a stronger association in pessimists than in optimists.

**Data Availability Statement:** Due to data security reasons (i.e., data contain potentially participant identifying information), the Heinz Nixdorf Recall

study does not allow sharing data as a public use file. The datasets used and/or analysed during the current study available from the corresponding author on reasonable request. Data requests can also be addressed to the data manager of the HNR study (recall@uk-essen).

**Funding:** None of the authors received funding for this publication.

**Competing interests:** The authors have declared that no competing interests exist.

**Abbreviations:** BDI, Beck Depression Inventory; CES-D, Center for Epidemiological Studies Depression Scale; DSM, Diagnostic and Statistical Manual of Mental Disorders; ICD, International Statistical Classification of Diseases and Related Health Problems; LCU, life change units; LOT-R, Life Orientation Test-Revised; SRRS, Social Readjustment Rating Scale.

## Conclusions

Experiencing life-changing events is associated with depression. Women and individuals with pessimistic personality are especially vulnerable which should be considered in prevention strategies.

## 1. Introduction

It has consistently been demonstrated that negative life events often precede the onset of major depressive episodes [1]. Complex study designs such as longitudinal analyses of population-based samples including dizygotic and monozygotic twins have made it possible to conclude that life events actually trigger, that is causally influence depressive reactions, instead of just being symptoms of depression [2].

Previous research investigating the association between life events and depression was mostly performed in case-control studies using fixed lists of life events. These studies showed that patients diagnosed with depression reported more life events and especially more undesirable life events than non-depressed controls from the general population [3], hospital staff [4], or patients with nonpsychiatric disease [5].

Fewer studies were performed in general population samples. These studies mostly compared depressive participants identified by scores above the cut-off in established depression screenings such as the Center for Epidemiologic Studies Depression Scale (CES-D) or Beck Depression Inventory (BDI) with non-depressive participants identified by scores below the cut-off in these depression screenings. So far, the largest population-based study on the cross-sectional association between life events and depression was performed within the Outcome of Depression in Europe Network. In 8,787 participants from 5 European countries, depressive participants (defined by BDI score >12), reported a higher number of life events (assessed with the List of Threatening Experiences including 12 negative life events) than non-depressive participants [6]. Also, in a smaller sample of the Kuopio Depression Study including 1,339 participants from the Finnish general population, depressive participants (defined as BDI score >9) reported a higher number of life events (assessed with a list of 12 negative life events similar to the List of Threatening Experiences) than non-depressive participants [7]. The minority of population-based studies assessed depression with structured diagnostic interviews because–compared with self-report screenings–structured diagnostic interviews require a lot more time and psychiatrically trained staff and thus are often not feasible in large population-based studies designed to address multiple research questions. Nonetheless, Assari and Lankarini assessed 12 months major depressive episode with a modified version of the World Mental Health Composite International Diagnostic Interview, a fully structured diagnostic interview, in a large population-based sample of American adults comprising 5,008 Blacks (African-Americans or Caribbean Blacks), and 891 Non-Hispanic Whites from the National Survey of American Life. They observed a significant association between the mean number of life events (assessed with a list of 8 stressful life events adequate for multiethnic samples) and major depressive episode independent of race [8]. Similarly, analyses of the Americans' Changing Lives study showed that the experience of an event from a list of 11 stressful life events was significantly associated with major depressive episode, also assessed with structured diagnostic interview in a population-based sample of 1,024 men and 1,800 women. When the life events were analyzed separately, especially death of a spouse or child, death of a friend or relative, and divorce or marital/love problem were significantly associated with major depressive episode [9].

Even though the above-mentioned studies showed that depressives more often report life events than non-depressed controls, not every individual who experiences major life events becomes depressed [10]. Research aiming to understand why some persons develop depression whereas others do not has directed attention to (a) whether life events differ in their stress-inducing properties and (b) whether life events have a differential impact across subgroups of individuals. Early studies could show that certain life events, such as death of a close person or change in relationships, are consistently judged to be more stressful than others [11, 12]. There is an ongoing debate about how to measure the impact of life events. The Social Readjustment Rating Scale (SRRS) developed by Holmes & Rahe (1967) is a widely used tool in research on the relationship between life events and various types of illness. The SRRS is based on the observation that life events per se are stressful because they require adjustment of an individual's life regardless of the desirability of the event [13]. Therefore, both desirable (positive) and undesirable (negative) life events are combined in the life stress-score, which has the advantage that it represents a continuous measure offering better statistical properties than presence/absence of short lists of life events. The SRRS has not been used in population-based studies on the association between life events and depression yet. So far, it has only been used in 3 studies with depressive patients [14–16]. Two of these studies including 90 major depressive patients and 121 controls, and 79 major depressive patients and 102 controls, respectively, showed that depressives had higher stress-scores than controls [14, 16]. One large study including 10,257 depressive patients observed higher stress-scores to be associated with a higher number of depressive episodes and depression severity [15].

There is still limited knowledge on moderators of the life events–depression association [8]. Various factors in addition to life events have been shown to increase the risk of depression and should be analyzed as possible moderating factors in life event research [1]. Despite considerable age- and sex differences in the prevalence of depression, only few studies analyzed the influence of sex [6, 8, 9] and age [17–19] on the association between life events and depression. Regarding the influence of sex, previous evidence is heterogenous with the analysis of Outcome of Depression in Europe Network data observing no significant interaction between sex and negative life events [6], the National Survey of American Life study stronger associations for men than for women [8] and the Americans' Changing Lives study stronger associations for women than for men [9]. Previous evidence mostly does not support a significant moderating effect of age on the association between number of life events or presence of specific life events and depression, except for the events of maternal loss or being unmarried, which put younger persons at higher risk [17–19]. Due to this scarce and heterogenous evidence on factors moderating the association between life events and depression, we decided to focus on age- and sex differences in the association between life events and depression and also analyze the influence of optimism, which has been shown to be associated with a more favorable way of coping with life events [20–22], leading to lower rates of depression [22]. Further, no population-based study has analyzed life events using a continuous life event stress-score yet. From a clinical point of view, subclinical depression is especially important because it is associated with an increased risk of developing major depression and because it already represents a condition with significant psychological difficulties and need for treatment. To close the gaps of knowledge, we quantified life events with the SRSS, which offers a continuous score of life-event stress and analyzed its relationship with depressive symptoms in a large random sample of the adult German population. In addition, we analyzed a possible effect modification by age, sex, and optimistic personality.

## 2. Methods

### 2.1 Participants

Data was drawn from the baseline examination of the Heinz Nixdorf Recall study, a prospective population-based study focusing on risk factors for cardiovascular disease and mortality. A random sample of men and women aged 45–75 years were enrolled via mandatory citizen registries in Essen, Bochum, and Mülheim/Ruhr between December 2000 and August 2003 and received 2 follow-up examinations after 5 and 10 years. The study design has been described in detail elsewhere [23]. The total cohort included 4814 participants (50.2% female) with a mean age of 59.6 years and a standard deviation of 7.8 years. The present analysis only uses the baseline data. The study was approved by the ethical committee of the University of Duisburg-Essen, Germany. All participants gave written informed consent and all methods were carried out in accordance with the relevant guidelines and regulations of the ethical committee and the 1964 Helsinki declaration and its later amendments.

### 2.2 Measures

**2.2.1 Social Readjustment Rating Scale (SRRS) for the assessment of life events and life event stress.**   Within a self-administered paper-pencil questionnaire, participants answered the question whether they had experienced any life events during the previous 6 months which were very important or life-changing (such as death, severe illness of a close person, career change, separation, move). For this, they had to tick a box for "No" or "Yes". If the participants answered that they had experienced an important or life-changing event during the previous 6 months, that is if they ticked "Yes", they were asked to describe the event in an open response format, meaning that they could freely write down these events on several lines. One rater evaluated all participant responses and scored them according to the Social Readjustment Rating Scale by Holmes and Rahe [11]. A small sample (n = 20 participants who reported life events) was rated by a second rater to determine interrater reliability. The SRRS represents the checklist approach and consists of 43 stressful life events generated based on clinical research to characterize the events that most often occurred to patients before seeking treatment. The SRRS is usually presented as a checklist and participants check these events if they have experienced them previously. However, the open response format used in our study offers more flexibility and is less time-consuming for the participants. Based on normative data, weights (life change units, LCUs) were assigned to each life event by the rater ranging from 11 to 100 to describe life event severity. Life event severity is characterized by the stress elicited by life events due to the required changes in usual activities (readjustment). In the process of the construction of the SRRS by Holmes and Rahe, the LCU for each event was determined by average ratings of a large group of subjects who rated all events regarding the amount of social readjustment that the experience of each life event required. The event "marriage" was assigned a value of 500 as an arbitrary anchor point, events needing more readjustment should be given higher ratings and events needing less readjustment lower ratings. Mean values were obtained for each event and divided by the constant of 10 to achieve a handy average amount of social readjustment required by the events. These values are termed LCUs and are summed up to a total life stress-score. LCUs of all life events experienced by each study participants were summed up to create a total score of life stress. Higher total scores represent a higher amount and duration of change in the participant's accustomed pattern of life resulting from various life events, regardless of their desirability. A high interrater reliability for the SRRS score was achieved (interrater correlation = 0.86).

**2.2.2 Life Orientation Test–Revised (LOT-R) for the assessment of optimism.** Optimism was assessed with the Life Orientation Test-Revised [24]. Optimism is regarded as a stable personality trait. Optimists expect things to go well and believe that future outcomes will be good rather than bad [25]. Optimism has been associated with a more favorable way of coping with negative life events [21, 22] and better adjustment towards important life transitions [20], leading to lower rates of depression and better overall mental and physical health [22]. The LOT-R consists of 10 items, 3 assessing optimism, 3 assessing pessimism and 4 fillers. The filler items were not included in the present study. Each item is scaled on a 5-point Likert scale with responses ranging from 0"strongly agree" to 4 "strongly disagree". After reversing the responses on the statements assessing pessimism, all responses to the 6 items are summed up. The total LOT-R score can range from 0 to 24 with higher values representing a higher disposition of optimism.

**2.2.3 Center for Epidemiological Studies Depression Scale (CES-D) for the assessment of depressive symptoms.** Depressive symptoms over the preceding week were assessed by self-administered questionnaire through the 15-item Center for Epidemiologic Studies Depression scale (CES-D) [26]. The CES-D is a screening tool for measuring depressive symptoms, which was specially designed for the use in non-clinical epidemiological populations but also validated in different psychiatric and psychosomatic clinical samples. Each item asks for feelings or behaviors during the previous week and is scaled on a 4-point scale with responses ranging from 0"rarely or none of the time (less than 1 day)" to 4 "most or all of the time (5–7 days)". Possible scores for the 15-item version range from 0 to 45, with higher levels indicating more or more frequent depressive symptoms. The CES-D is considered an indicator of a probable depressive episode but is not equivalent to a face-to-face physician diagnosis. The assessment of signs of depression is particularly important, since they show a higher prevalence compared to the clinical depression diagnosis according to Diagnostic and Statistical Manual of Mental Disorders (DSM) or International Statistical Classification of Diseases and Related Health Problems (ICD). Further, signs of depression represent a frequent comorbidity and exert an important influence on the recovery from somatic and psychosomatic diseases.

### 2.3 Statistical analysis

Continuous data are presented as mean (standard deviation) for normally distributed or median (Q1,Q3) for non-normally distributed data, categorical data are shown as number (%). Statistical comparisons between two independent groups (e.g., men vs women), were done by student's t-test for normally distributed continuous data, by Mann-Whitney test for non-normally distributed continuous data, and by Chi-square test or Fisher's exact test for categorical data. Associations between life events (categorical as yes vs no or—in case participants reported important life events -, as SRRS-Score >Q3 vs ≤Q3) and depressive symptoms (continuous CES-D score) were analyzed by unadjusted linear regression analyses and multivariable linear regression analyses adjusted for age (>65 vs ≤65 years) and sex (male vs female, model 1), and additionally for optimism (LOT-R-Score >Q3 vs ≤Q3, model 2). Interactions between life events and adjusting variables were analyzed as multiplicative interactions. All hypothesis tests used two-sided tests, and p-values < .05 were considered statistically significant. Missing values were excluded listwise. All analyses were done with IBM SPSS Statistics 21 for Windows (IBM Corporation, Armonk, NY, USA).

## 3. Results

### 3.1 Presence of important life events

Valid data on the presence of important or life-changing events during the previous 6 months was available for 4,664 (96.9%) of the total cohort of 4,814 participants. Of those 4,664

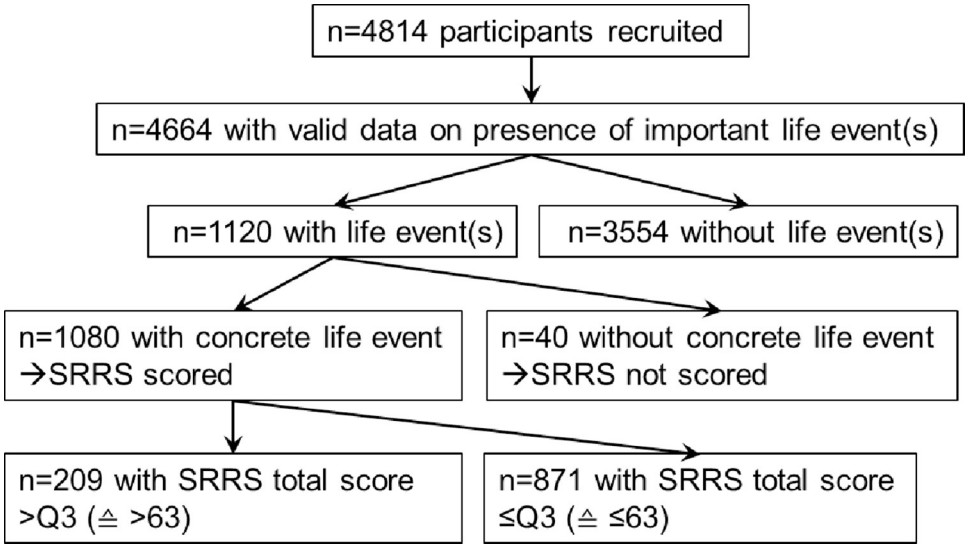

**Fig 1. Flowchart of the Heinz-Nixdorf Recall analyses samples.** SRRS, Social Readjustment Rating Scale.

participants, 1,120 (24.0%) reported that they had experienced such an important life event. Among those 1,120 participants who experienced an important life event during the previous 6 months, 1,080 (96.4%) recalled one or more concrete events in the subsequent free response format (Fig 1).

Most of those 1,080 participants recalled only one important life event which was listed in the SRRS (n = 925, 85.6%), 135 participants (12.5%) recalled two events, and only very few participants reported three (n = 18, 1.7%) or four events (n = 2, 0.2%). Change in health of a family member (n = 310, 24.7%), death of a close family member, (n = 210, 16.7%), and personal injury or illness (n = 124, 9.9%) were the most frequent of all reported life events (Table 1).

### 3.2 Life event stress

As explained before, the SRRS offers the possibility to create a total score (stress-score), with higher scores representing a higher amount and duration of change in the participant's accustomed pattern of life resulting from the experience of various life events. The stress-score was not normally distributed but right-skewed, most participants had low scores (S1 Fig in S1 File). Of the 1080 participants, who recalled at least one concrete life event during the previous 6 months, the median stress-score was 45.0 (Q1 = 39.0, Q3 = 63.0), the minimum was 20.0, the maximum 182.0. Stress-scores >Q3 were defined as high life event stress, stress-scores ≤Q3 as low life event stress.

### 3.3 Optimism

Valid data on the personality trait of optimism assessed via the sum score of the LOT-R was available for 4,687 (97.4%) of the total cohort of 4,814 participants. Total LOT-R scores were not normally distributed but slightly skewed to the left with only few participants exhibiting low optimism scores (S2 Fig in S1 File). The median LOT-R score was 15.0 (Q1 = 13.0, Q3 = 18.0), the minimum was 0.0, the maximum 24.0. LOT-R scores >Q3 were defined as optimism, LOT-R scores ≤Q3 as pessimism.

**Table 1. Frequency of important life events according to the SRRS.**

| Life event | Life Change Unit | Number (%) |
|---|---|---|
| Change in health of family member | 44 | 310 (24.7) |
| Death of close family member | 63 | 210 (16.7) |
| Personal injury or illness | 53 | 124 (9.9) |
| Change in work hours or conditions | 20 | 73 (5.8) |
| Change in residence | 20 | 69 (5.5) |
| Death of close friend | 37 | 59 (4.7) |
| Retirement | 45 | 56 (4.5) |
| Change in living conditions | 25 | 42 (3.3) |
| Death of spouse | 100 | 38 (3.0) |
| Fired at work | 47 | 36 (2.9) |
| Marital separation | 65 | 33 (2.6) |
| Change in financial state | 38 | 31 (2.5) |
| Gain of new family member | 39 | 30 (2.4) |
| Change in number of arguments with spouse | 35 | 26 (2.1) |
| Change to different line of work | 36 | 25 (2.0) |
| Trouble with in-laws | 29 | 25 (2.0) |
| Divorce | 73 | 22 (1.8) |
| Son or daughter leaving home | 29 | 22 (1.8) |
| Change in responsibilities at work | 29 | 9 (0.7) |
| Trouble with boss | 23 | 5 (0.4) |
| Wife begin or stop work | 26 | 3 (0.2) |
| Marriage | 50 | 2 (0.2) |
| Revision of personal habits | 24 | 2 (0.2) |
| Jail term | 63 | 1 (0.1) |
| Marital reconciliation | 45 | 1 (0.1) |
| Mortgage over $10,000 | 31 | 1 (0.1) |
| Outstanding personal achievement | 28 | 1 (0.1) |
| Change in number of family get-togethers | 15 | 1 (0.1) |

SRRS, Social Readjustment Rating Scale. Frequencies shown in descending order.

### 3.4 Depressive symptoms

Valid data on depressive symptoms assessed via the total score of the 15-item version of the CES-D was available for 4,645 (96.5%) of the total cohort of 4,814 participants. Total CES-D scores were not normally distributed but right-skewed, most participants had low depression scores (S3 Fig in S1 File). The median depression score was 7.0 (Q1 = 4.0, Q3 = 11.0). The minimum depression score was 0.0, the maximum 42.0.

### 3.5 Association between presence of important life events and depressive symptoms

Participants who reported that they had experienced an important life event during the previous 6 months (n = 1,120) had significantly higher depression scores (Median = 8, Q1 = 5, Q3 = 14) than participants who did not report the experience of an important life event during the previous 6 months (n = 3546, Median = 6, Q1 = 3, Q3 = 10, p < .001; Fig 2).

In an unadjusted linear regression, the presence of important life events was associated with a significant increase in CES-D score (B = 2.6, 95%CI = 2.2 to 3.0, p < .001). In

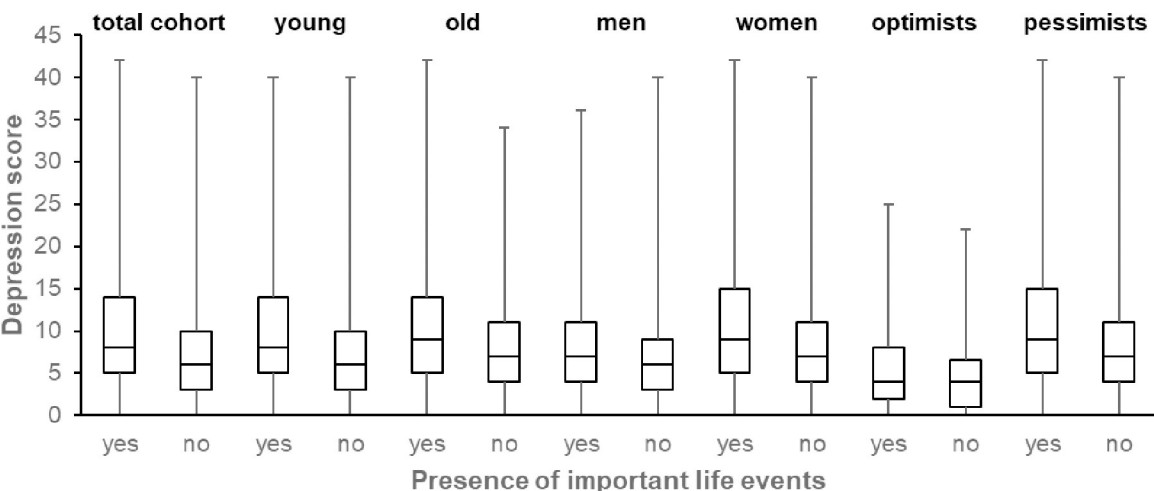

**Fig 2. Box and whisker plots demonstrating Center for Epidemiological Studies Depression Scale (CES-D) depression score for presence and absence of important life events in the total cohort and stratified by age (young ≤65 years vs old >65 years), sex (men vs women), and optimism (optimists Life Orientation Test-Revised (LOT-R) score >18 vs pessimists LOT-R score ≤18).** The horizontal line in each box represents the median with the box representing the interquartile range (Q3-Q1) and the whiskers representing the total range of the data (max-min).

multivariable regressions adjusted for age and sex as well as age, sex, and optimism, presence of important life events remained significantly associated with higher depression score (Table 2). Presence of important life events, age, sex, and optimism explained 10.8% of the total variation in depression score.

## 3.6 Interacting influence of the presence of important life events with age, sex, and optimism on depressive symptoms

The interaction analyses of the presence of important life events with the adjusting variables age, sex, and optimism on depressive symptoms showed a significant interaction between life events and sex (B = -0.8, 95%CI = -1.6 to -0.1, p = .043) and between life events and optimism (B = -0.7, 95%CI = -0.8 to -0.6, p < .001) in the fully adjusted model 2. These interactions revealed the stronger influence of the presence of life events on women compared to men and on pessimists compared to optimists.

**Table 2. Association between presence of important life events (yes vs no) and depressive symptoms.**

|  | Unadjusted | | | Model 1 | | | Model 2 | | |
|---|---|---|---|---|---|---|---|---|---|
|  | **B** | **95% CI** | **P** | **B** | **95% CI** | **P** | **B** | **95% CI** | **p** |
| Total cohort | 2.6 | 2.2 to 3.0 | < .001 | 2.6 | 2.2 to 3.0 | < .001 | 2.5 | 2.1 to 2.9 | < .001 |
| >65 years | 2.7 | 1.9 to 3.6 | < .001 | 2.6 | 2.1 to 3.0 | < .001 | 2.6 | 1.8 to 3.4 | < .001 |
| ≤65 years | 2.7 | 2.2 to 3.1 | < .001 | 2.6 | 1.8 to 3.5 | < .001 | 2.5 | 2.0 to 2.9 | < .001 |
| Men | 1.9 | 1.4 to 2.5 | < .001 | 1.9 | 1.4 to 2.5 | < .001 | 1.9 | 1.3 to 2.4 | < .001 |
| Women | 3.0 | 2.4 to 3.6 | < .001 | 3.1 | 2.5 to 3.7 | < .001 | 3.1 | 2.5 to 3.7 | < .001 |
| Optimists | 1.2 | 0.4 to 1.9 | .002 | 1.1 | 0.4 to 1.8 | .002 | 1.1 | 0.4 to 1.8 | .002 |
| Pessimists | 2.9 | 2.4 to 3.3 | < .001 | 2.8 | 2.3 to 3.2 | < .001 | 2.8 | 2.3 to 3.2 | < .001 |

B = unstandardized regression weights from linear regression analysis, CI = confidence interval, Model 1 adjusted for age (>65 vs ≤65 years) and sex (male vs female), Model 2 adjusted for age, sex, and optimism (score >Q3 vs ≤Q3); in the stratified analyses not adjusted for the stratification variable.

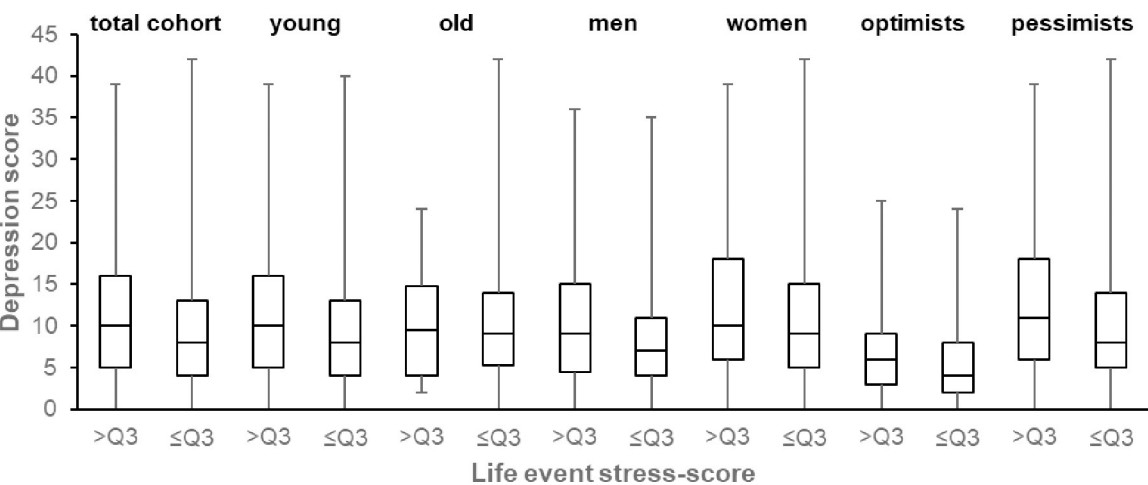

**Fig 3. Box and whisker plots demonstrating Center for Epidemiological Studies Depression Scale (CES-D) depression score for Social Readjustment Rating Scale (SRRS) total score >Q3 and ≤Q3 (>63 vs ≤63), representing high and low life event stress.** Values are shown for the total cohort and stratified by age (young ≤65 years vs old >65 years), sex (men vs women) and optimism (optimists Life Orientation Test-Revised (LOT-R) score >18 vs pessimists LOT-R score ≤18). The horizontal line in each box represents the median with the box representing the interquartile range (Q3-Q1) and the whiskers representing the total range of the data (max-min).

In absolute numbers, men who reported to have experienced an important life event during the previous 6 months (21.2%), had significantly higher depression scores (Median = 7, Q1 = 4, Q3 = 11) than men who reported not having experienced such an event (78.8%, Median = 6, Q1 = 3, Q3 = 9, p < .001). However, this difference was more pronounced for women: women who reported to have experienced an important life event during the previous 6 months (n = 26.8%), had significantly higher depression scores (Median = 9, Q1 = 5, Q3 = 15) than women who reported not having experienced such an event (73.2%, Median = 7, Q1 = 4, Q3 = 11, p < .001, Fig 2).

Optimists, who reported to have experienced an important life event during the previous 6 months (22.4%), had significantly higher depression scores (Median = 4, Q1 = 2, Q3 = 8) than optimists who reported not having experienced such an event (77.6%, Median = 4, Q1 = 1, Q3 = 7.0, p = .005). This difference was more pronounced for pessimists: pessimists who reported to have experienced an important life event during the previous 6 months (24.4%), had significantly higher depression scores (Median = 9, Q1 = 5, Q3 = 15) than pessimists who

**Table 3. Association between life event stress (score >Q3 vs ≤Q3) and depressive symptoms.**

| | Unadjusted | | | Model 1 | | | Model 2 | | |
|---|---|---|---|---|---|---|---|---|---|
| | **B** | **95% CI** | **p** | **B** | **95% CI** | **p** | **B** | **95% CI** | **p** |
| Total cohort | 2.2 | 1.1 to 3.3 | < .001 | 1.9 | 0.8 to 3.1 | < .001 | 1.9 | 0.8 to 2.9 | < .001 |
| >65 years | -0.3 | -3.3 to 2.8 | .87 | -1.2 | -4.2 to 1.8 | .42 | -1.3 | -4.2 to 1.6 | 0.39 |
| ≤65 years | 2.6 | 1.4 to 3.8 | < .001 | 2.4 | 1.2 to 3.6 | < .001 | 2.3 | 1.1 to 3.4 | < .001 |
| Men | 2.2 | 0.6 to 3.9 | .007 | 2.2 | 0.6 to 3.8 | .007 | 2.2 | 0.6 to 3.8 | .006 |
| Women | 1.6 | 0.1 to 3.1 | .032 | 1.8 | 0.3 to 3.3 | .022 | 1.6 | 0.2 to 3.1 | .029 |
| Optimists | 0.9 | -1.0 to 2.9 | .34 | 0.9 | -1.0 to 2.9 | .34 | 0.9 | -1.0 to 2.9 | .34 |
| Pessimists | 2.3 | 1.1 to 3.6 | < .001 | 2.0 | 0.8 to 3.2 | .001 | 2.0 | 0.8 to 3.2 | .001 |

B = unstandardized regression weights from linear regression analysis, CI = confidence interval, Model 1 adjusted for age (>65 vs ≤65 years) and sex (male vs female), Model 2 adjusted for age, sex, and optimism (score >Q3 vs ≤Q3); in the stratified analyses not adjusted for the stratification variable.

reported not having experienced such an event (75.6%, Median = 7, Q1 = 4, Q3 = 11, p < .001, Fig 2).

Presence of an important life event and depression scores stratified by age, sex, and optimism are shown in S1 Table in S1 File. Presence of all important life events of the SRRS stratified by age, sex, and optimism are shown in S2-S4 Tables in S1 File.

### 3.7 Association between life event stress and depressive symptoms

Among the 1,120 participants who reported to have experienced an important life event during the previous 6 months, 1,080 reported a concrete life event in the free response format (Fig 1). These concrete life events were scored according to the SRRS [11] and a stress-score was created to describe the readjustment of each individual's life, which resulted from the experience of these life events.

Participants with high life event stress, defined as stress-score >Q3 (n = 290) had significantly higher depression scores (Median = 10, Q1 = 5, Q3 = 16) than participants with low life event stress, defined as stress-score ≤Q3 (n = 871, Median = 8, Q1 = 4, Q3 = 13, p < .001, Fig 3).

In an unadjusted linear regression, high life event stress was associated with a significant increase of depression score (B = 2.2, 95%CI = 1.1 to 3.3, p < .001). Even in multivariable regressions adjusted for age and sex as well as age, sex, and optimism, high life event stress was significantly associated with higher depression score (Table 3). Life event stress, age, sex, and optimism explained 10.2% of the total variation in depression score.

### 3.8 Interacting influence of life event stress with age, sex, and optimism on depressive symptoms

The interaction analyses of life event stress with the adjusting variables age, sex, and optimism on depressive symptoms showed a significant interaction between life event stress and optimism (B = -1.0, 95%CI = -1.2 to -0.7, p < .001) in the fully adjusted model 2. This interaction revealed the stronger influence of life event stress on pessimists compared to optimists.

In absolute numbers, optimists with high life event stress (n = 18.7%), had significantly higher depression scores (Median = 6, Q1 = 3, Q3 = 9) than optimists with low life event stress (81.3%, Median = 4, Q1 = 2, Q3 = 8, p < .001). This difference was more pronounced for pessimists: pessimists with high life event stress (19.7%), had significantly higher depression scores (Median = 11, Q1 = 6, Q3 = 18) than pessimists with low life event stress (80.3%, Median = 8, Q1 = 5, Q3 = 15, p < .001, Fig 3).

Life event stress and depression scores stratified by age, sex, and optimism are shown in S1 Table in S1 File.

Exclusion of participants with a history of coronary heart disease (n = 327) did not change our results to a relevant degree (not shown).

## 4. Discussion

### 4.1 Principal findings

For the first time, we observed a significant association between the presence of important life events and depressive symptoms in the middle-aged to old-aged general population. In participants who experienced important life events, higher life event stress-scores due to the necessary life adjustment after important life events were significantly associated with higher depression scores. Women and individuals with pessimistic personality were more susceptible

to the adverse influence of life events on depression than men and individuals with optimistic personality.

## 4.2 Comparison with other studies

Presence of important life events, life event stress and their association with depressive symptoms have not been assessed in the general adult population before. So far, the largest population-based study on the association between life events and depression was performed within the Outcome of Depression in Europe Network [6]. In 8,787 participants from 5 European countries, presence of 12 negative life events during the previous 6 months, included in the List of Threatening Experiences, and depression, defined by BDI score >12, were assessed. A higher number of negative life events was significantly associated with increased rates of depression. In contrast to our study, there was no significant interaction between sex and negative life events with respect to depression. Like in our study, illness of a relative and death of close friends/relatives were experienced most often. While we focused on the idea that not only negative, but also positive life events can have an influence on an individual's emotional state, because positive events can require changes to an individual's life which are experienced as stressful, this study only focused on negative life events. Further, it did not assess the stress-potential of each life event like we did, but only analyzed the number of negative life events as 0, 1, 2, ≥3. Also, in a smaller study including 1,339 participants from the Finnish general population, life events were assessed with a list of negative life events similar to the List of Threatening Experiences and depression with BDI score but with a cut-off of 9 [7]. Again, only the number of life events was analyzed with the mean number of life events in the depressive group being significantly higher than in the non-depressive group. Using a structured diagnostic interview to assess depression, the large population-based National Survey of American Life comprising 5,008 Blacks and 891 Non-Hispanic Whites observed a significant association between the mean number of life events (assessed with a list of 8 stressful life events adequate for multiethnic samples during the previous 30 days) and major depressive episodes during the previous year [8]. In contrast to our study, the association between stressful life events and major depressive episodes was stronger for men than for women. Reasons for the divergent results might be 1.) the less comprehensive assessment of life events with a list of only 8 events compared to the free response format in our study and coding according to the SRRS which includes 43 life events, and 2.) the assessment of clinical major depressive episodes compared to our assessment of subclinical depressive symptoms which was developed for the use in population-based studies. Men tend to report different life events than women (S3 Table in S1 File) and as operationalized in the SRRS, different life events have different stress-inducing properties having a different probability to induce clinical vs subclinical depression. Moreover, specific life events are more adverse for one sex than another, i.e., job loss is usually more severe for men than for women while loss of a family member is usually more severe for women than for men [9]. With short life event checklists, there is a higher probability of having an overrepresentation of life events more often experienced by one sex or overrepresentation of life events leading to clinical vs subclinical depression. The population-based Americans' Changing Lives study including 1,024 men and 1,800 women also showed a significant association between major depressive episode, again assessed with structured diagnostic interview and the experience of an event from a list of 11 stressful life events [9]. In line with our results, this association was stronger for women than for men. Regarding single life events, death of a friend or relative was experienced most often, confirming the observations of our study. In contrast to our study, life event stress was not examined.

To the best of our knowledge, previous evidence on the moderating effect of age on the association between life event stress and depression is scarce. In our study, age did not moderate the association between life events and depression. Subjects ≤65 years reported the presence of an important life event more often than subjects >65 years, however, life event stress and severity of depressive symptoms were not significantly associated with age. Life course epidemiology suggests that life events can have differential effects on depression, depending on the age when they occur. Therefore, future longitudinal studies should investigate age effects in more depth [27]. Previous studies with depressive and non-depressive subjects observed that younger participants reported more life events than older participants and that prevalence of depression decreases in older age [17] One Danish study, which used registry data to assess prevalence of specific life events (vital, marital and employment status) instead of directly asking participants, similar to our results observed no interaction between most of the life events assessed and age regarding the outcome of being admitted for the first time ever at a psychiatric hospital and discharged with a diagnosis of depression. Only for the life event "being unmarried", a significant interaction was observed with younger persons having a higher risk to become diagnosed with depression [17]. Also a smaller case-control study including 64 depressive and 74 non-depressive patients observed no significant interaction between age and the number of reported life events [18]. For the specific category of loss events, one larger epidemiological study including 3,491 healthy individuals recruited through general practices, found the impact of maternal loss on the risk of developing depression to vary significantly by age, being highest in those younger at the time of loss, but no significant interaction for marital loss [19].

So far, only one smaller population-based Chinese study including 1,147 Hong Kong residents analyzed the influence of coping on the association between life events and mental health, but not specifically depression [28]. This study showed that resilience moderated the association between multiple traumatic life events and mental health (assessed by Short-Form 12 Health Survey). Resilient people were less susceptible to the negative influence of traumatic life events on mental well-being. This observation supports our results of a stronger influence of life events on individuals with pessimistic personality. Resilient people are characterized by optimism, positive coping and hardiness and can therefore be more flexible to cope with life challenges and adapt to life-changing events.

The SRRS [11] to assess life event stress caused by necessary adaptions of the previous life after the experience of important life events was so far applied only 3 times in the context of depression. In 2 case-control studies with 90 major depressive patients and 121 controls, and 79 major depressive patients and 102 controls, respectively, depressives had higher stress-scores than controls [14, 16]. In the smaller case-control study, stress-scores were not significantly associated with depression in multivariable logistic regression models [16]. In a large patient study comprising 10,257 patients with current single or recurrent major depressive episode, higher stress-scores were associated with a higher number of depressive episodes and depression severity [15].

### 4.3 Limitations

The design of our study possesses strengths and limitations. Important strengths are its large sample size and that it is representative of the general adult German population. Further, life events were comprehensively analyzed with the SRRS and depressive symptoms assessed with the CES-D representing a validated instrument specifically designed for the use in epidemiological studies. An important limitation is the cross-sectional design of the analysis, which does not allow to draw causal conclusions. The associations between life events and depression

could be bidirectional, since not only the experience of life events can lead to depression but also depression can lead to the experience of negative life events. Furthermore, the retrospective recall of life events may be biased by the current emotional state or defense mechanisms. Due to the open response format, it is possible that women/men and optimists/pessimists used different criteria to determine what constituted an important life event. Other variables not included in the present study such as chronic stress could contribute to the experience of life events or influence their association with depression.

### 4.4 Clinical implications

Even if our results are influenced by reverse causality, i.e., depression leading to specific life events, this still has important clinical implications in a way that the experience of life events must be considered more in the prevention and treatment of depressive episodes. Identification of stressful life events and working on strategies to prevent recurrent depressive episodes resulting from the vicious cycle of depression and stressful life events represents an important part of anti-depressive treatment [29]. A study in depressive adolescents could already show that standard therapy was not able to reduce depressive symptoms in the presence of high life event stress. Thus, the authors suggested that adolescents suffering from high life event stress might require specialized therapy [30]. In line with this idea, anti-depressive therapy, which was able to reduce the experience of negative life events, led to long-term improvement in psychological well-being in adult depressive patients [31].

### 5. Conclusions

This is the first study exploring the presence of a comprehensive selection of life events as well as the stress caused by the necessary adaptation of the previous life due to the experience of these life events in a large population-based study. Our observation that individuals who experienced important life events and high levels of life event stress had higher levels of depressive symptoms and that women and persons with pessimistic personality are especially vulnerable has important clinical implications. Our results implicate, that the occurrence of major life events signals a period of increased risk for new onset depression and recurrent depressive episodes, which has to be considered in preventing the evolution of distress to disorder and in treating depressive disorders. Prevention programs should be developed to help individuals suffering from stressful life events to better cope with the life-changing situation. Women and persons with a pessimistic outlook on life should be especially targeted with preventive therapies.

### Supporting information

**S1 File.**
(DOCX)

### Author Contributions

**Conceptualization:** Johannes Siegrist, Raimund Erbel, K-H. Jöckel.

**Formal analysis:** Janine Gronewold, Ela-Emsal Duman.

**Writing – original draft:** Janine Gronewold.

**Writing – review & editing:** Janine Gronewold, Ela-Emsal Duman, Miriam Engel, Miriam Engels, Johannes Siegrist, Raimund Erbel, K-H. Jöckel, Dirk M. Hermann.

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
