## [Decision Letter · Decision Letter 0]

13 Dec 2021

PONE-D-21-02934

Association between life events and later depression in the population-based Heinz Nixdorf Recall study – the role of sex and optimism

PLOS ONE

Dear Dr. Gronewold,

Thank you for submitting your manuscript to PLOS ONE. After careful consideration, we feel that it has merit but does not fully meet PLOS ONE’s publication criteria as it currently stands. Therefore, we invite you to submit a revised version of the manuscript that addresses the points raised during the review process.   I plus one reviewer 

have read your manuscript. We feel that it addresses an interesting question but all sections of the manuscript need to be revised before it can be accepted. In particular the introduction does not provide a sufficient review of previous related research discussing the relationship between depression and life  event experiences. Additional relevant research needs to be described and evaluated in more depth. Reference is also made to the stress arising from the COVID pandemic yet this paper does not address the impact of COVID. This point needs to be made clear or reference to COVID deleted. There is also insufficient justification for examining age as a moderator. The introduction mentions a lack of research on gender effects but the discussion mentions research that addresses this issue. Material in the discussion should also be in the introduction to create a better rationale for the study. See the comments of reviewer one for specific examples. The methods section lacks detail on numerous places. It is not clear when the study took place or how the experience of life events was measured. In the results section the difference on SRRS  and CED scores could analysed for the different subgroups. There are a number of issues concerning the discussion. The stance on causality  between life events and depression is not clear either in introduction or the discussion, but was particularly problematic  The discussion of  related findings is not well-organized, making it difficult to understand why the studies are grouped together. Additional points have been raised by Reviewer1. In your revision address all points raised by Reviewer 1 and the points outlined in this letter.

Please submit your revised manuscript by January 27, 2022. If you will need more time than this to complete your revisions, please reply to this message or contact the journal office at plosone@plos.org. Please include the following items when submitting your revised manuscript:

We look forward to receiving your revised manuscript.

Kind regards,

Barbara Dritschel, PhD

Academic Editor

PLOS ONE

Journal Requirements:

4. One of the noted authors is a group or consortium Heinz Nixdorf Recall Study Investigative Group. In addition to naming the author group, please list the individual authors and affiliations within this group in the acknowledgments section of your manuscript. Please also indicate clearly a lead author for this group along with a contact email address.

5. Your abstract cannot contain citations. Please only include citations in the body text of the manuscript, and ensure that they remain in ascending numerical order on first mention.

Reviewers' comments:

Reviewer's Responses to Questions

**Comments to the Author**

1. Is the manuscript technically sound, and do the data support the conclusions?

Reviewer #1: Partly

2. Has the statistical analysis been performed appropriately and rigorously? 

Reviewer #1: Yes

3. Have the authors made all data underlying the findings in their manuscript fully available?

Reviewer #1: No

4. Is the manuscript presented in an intelligible fashion and written in standard English?

Reviewer #1: Yes

5. Review Comments to the Author

Reviewer #1: This paper explored the association between the experience of important life events in the previous six months (measured using the Social Readjustment Rating Scale; SRRS) and depressive symptoms in a large population-based sample. In particular, the study examined whether sex and optimism moderated the association between life events and depression. The results showed that (1) people who had experienced important life events in the previous 6 months exhibited a greater number of depressive symptoms than people who had not experienced an important life event in the previous 6 months; (2) in those who had experienced an important life event, people with high SRRS scores exhibited a greater number of depressive symptoms than those with lower scores; (3) the association between SRRS score and depressive symptoms was stronger in women than men; and (4) the association between SRRS score and depressive symptoms was higher in pessimists than optimists. The authors concluded that women and pessimists should be targeted by prevention programmes to help them cope with stressful life events, and that specialized therapy may be required.

Comments/questions

1. The introduction opens with a reference to ongoing COVID pandemic as an example of an important, life-changing event (lines 47-48), which sets the study up as though it will measure the effect of the pandemic on depressive symptoms. However, it appears that the data for this study were in fact collected many years before the pandemic began. It would be good to make clearer in the introduction that this study is not about the pandemic specifically.

2. In general, the introduction does not describe the context for the study in enough detail. Information about previous studies that have examined the association between life events and depression is scarce (refs 5-9 on lines 59-62), giving the impression that there is relatively little existing research in this area. For example, the authors state that previous studies have not generally examined the effect of sex on this relationship (lines 78-80). However, in the discussion, two sex effects in previous studies are mentioned for the first time (lines 332-334 and lines 341-342), and several more details about these and other previous studies are presented. These details were needed in the introduction, to make it clear how the current study builds on previous work.

3. In lines 78-81 and line 90, age is mentioned as a potential moderating factor and a main focus of this study. However, although age was included in the models, it was not discussed in any detail.

4. In the method section the authors state that the data for this study were drawn from a previous study for which participants were recruited around 20 years ago, with follow-ups at 5 and 10 years (lines 94-98). It is not clear when the data for this study were actually collected – at the outset of the original study, in a follow-up examination, or across multiple time points?

5. It is not entirely clear what instructions participants were given for the self-administered questionnaire (lines 104-108). Were the examples of important life events on line 106 provided to participants as examples, or were the participants left to decide on their own what constituted an important life event (this seems to be what is implied by “open response format” on line 108/113-114)? If left to decide on their own, is it possible that women/men and optimists/pessimists used different criteria to determine what constituted an important life event?

6. On lines 112-115 the authors argue that their open response format offers more flexibility than using a checklist of life events, yet participants’ responses were evaluated according to a checklist of events for which normative data on severity were available. It is not clear why the open response format is therefore more flexible. Perhaps the authors could expand on this idea. To what extent were participants left to self-determine what constituted an important life event?

7. Was there a reason why all of the life events were evaluated by only one rater (lines 108-110)? How can the authors be sure that the rater’s evaluations were reliable? Given the flexible open-response format, did any participants report life events that were not on the existing checklist of events? If so, how were these responses dealt with?

8. Since the comparisons of interest in this study were women vs. men and optimists vs. pessimists, it would have been helpful to see some data on the SRRS scores and CES-D scores in these different subgroups. Were optimists and pessimists equally likely to report important life events, for example?

9. The argument is made that both positive and negative life events can be experienced as stressful due to the need to readjust (e.g., lines 72-73, lines 126-127). Following on from point 6 above, is it possible that, for example, optimists were more likely to report positive important life events, and this could explain the weaker relationship with depressive symptoms?

10. In the abstract, the authors state that they examine the presence of important life events and the RESULTING life event stress (line 26). Similarly, elsewhere the argument is made that higher stress is associated with higher depression scores (e.g., line 310-312). However, since the study did not directly measure stress, it is not clear that the participants in this study experienced stress, only that these life events tend to be stressful. These claims could be worded a little more carefully, to reflect what was actually measured.

11. The authors’ stance on the causality of the relationship between life events and depression was unclear. In lines 52-55 of the introduction, they state that a robust and causal association between stressful life events and depression has become established, whereby stressful life events trigger depression. Later, it is acknowledged that the current study is unable to draw conclusions about causality, and that the authors rely heavily on other studies to conclude that stressful life events cause depression (line 376). The arguments in the conclusion then appear to depend on the causality evidenced by other studies, rather than on the data presented in the current study. Given that this study is apparently the first to measure the moderating effects of sex and optimism, how can we be confident that this assumption of causality is justified?

12. In the discussion section, there is a list (beginning on line 315) of several previous studies that have explored similar research questions, albeit using different scales, sampling methods, etc. Some of the findings of these previous studies were consistent with the current findings, while others were inconsistent. The list format is quite repetitive, and it is difficult to follow the argument throughout – what is the overall message? The argument would be clearer if the findings from the previous studies could be aggregated in a meaningful way, such as by the methods used, the time period sampled, etc.

13. It is not clear why a less comprehensive assessment of life events using a checklist of 8 events vs. free response would potentially reverse the sex effect (lines 334-335). This argument needs to be made more explicit. The same applies to the potential difference between clinical and subclinical depression (lines 336-338).

14. On lines 356-363, there is a section on previous uses of SRRS (refs 11, 12, 13) showing that depressives had higher stress scores (i.e., more/more serious life events) than controls in 2/3 studies, and that stress score associated with depression severity in one study. This information would be better in the introduction (see point 2, above).

15. On lines 378-380, the authors argue that even if the causality is reversed, such that depression leads to the experience of stressful life events, “this still has important clinical implications in a way that the experience of life events must be considered more in the prevention and treatment of depression”. This argument was difficult to follow; if depression causes stressful life events, how could consideration of the experience of life events prevent or treat depression?

16. A new study is introduced in the conclusion on lines 389-396. It would be better to include this argument in the main body of the discussion so that the conclusion remains focused on the current investigation.

Minor points

Line 84-85 reads “From the clinical point of view, especially subclinical depression is important because it is associated with…”; it would read better as “From a clinical point of view, subclinical depression is especially important because….”

Lines 152-155 – this sentence is quite long, and would read better with commas.

Lines 310-311 – “In case important life events were experienced” would read better as “In participants who experienced important life events”

Line 311-312 – “higher stress… was again significantly associated…” – the “again” appears to be redundant here

Line 313 – “Women and individuals with pessimistic personality were more susceptible to the negative influence of life events on depression”. The wording is ambiguous, as it seems to suggest there is a negative association between life events and depression, rather than the positive association that was observed.

Line 317 – “The so far largest population-based study” would read better as “So far, the largest population-based study”

Line 350 – “this study could show….” presumably means “this study showed that….”?

6. PLOS authors have the option to publish the peer review history of their article (what does this mean?). If published, this will include your full peer review and any attached files.

Reviewer #1: No

---

## [Author Response · Author response to Decision Letter 0]

7 Mar 2022

Gronewold J, et al.

‘Association between life events and later depression in the population-based Heinz Nixdorf Recall study – the role of sex and optimism’ (MS ID: PONE-D-21-02934) 

Responses to editor and reviewers

Please note that all new/revised text is highlighted in yellow in the manuscript.

Editor

I plus one reviewer have read your manuscript. We feel that it addresses an interesting question but all sections of the manuscript need to be revised before it can be accepted. In particular the introduction does not provide a sufficient review of previous related research discussing the relationship between depression and life event experiences. 

We would like to thank the editor and reviewer for the critical evaluation and appreciation. We added a more in-depth revision of previous evidence in the introduction and discussion. 

Reference is also made to the stress arising from the COVID pandemic yet this paper does not address the impact of COVID. This point needs to be made clear or reference to COVID deleted.

We deleted the reference to COVID. 

There is also insufficient justification for examining age as a moderator.

In addition to depression risk, also the experience of life events changes throughout life. Previous evidence regarding the moderating effect of age on the association between life event and depression is scarce and showed heterogeneous results with some studies observing no significant moderation while others observed moderation effects for specific life events such as maternal loss or being unmarried being more harmful at younger ages. We now added this information to the introduction and discussion section.

The introduction mentions a lack of research on gender effects but the discussion mentions research that addresses this issue.

Research on the influence of sex on the association between life events and depression is scarce and completely lacking for life event stress. We now clarified the current state of research in our revised introduction.

Material in the discussion should also be in the introduction to create a better rationale for the study

As stated before, we added a more in-depth review of previous evidence in the introduction and discussion. 

The methods section lacks detail on numerous places. It is not clear when the study took place or how the experience of life events was measured.

The study is a cross-sectional analysis of baseline data of the Heinz Nixdorf Recall study, which was collected from 2000 to 2003. We added this information to the methods section and also added more details regarding the measurement of the experience of life events. 

In the results section the difference on SRRS and CED scores could be analysed for the different subgroups.

We now show these results in Supplemental Table S1 and also show presence of single life events stratified by age, sex, and optimism in Supplemental Tables S2-4. 

There are a number of issues concerning the discussion. The stance on causality between life events and depression is not clear either in introduction or the discussion, but was particularly problematic. 

We now clarified this issue in our revised introduction and discussion: previous studies with more complex study designs like longitudinal analyses of population-based samples including dizygotic and monozygotic twins have made it possible to conclude that life events actually trigger, that is causally influence depressive reactions instead of just being symptoms of depression. However, these studies did not include continuous scores of life event stress and did not analyze moderators of the associations between life events and depression. We closed this gap of knowledge in our present study, but since our study is a cross-sectional observational study, we clarified that it is not suited to draw causal conclusions in our revised limitations section.

The discussion of related findings is not well-organized, making it difficult to understand why the studies are grouped together.

We reorganized our discussion and grouped studies by methods used to assess life events and depression. It now begins with:

Dalgard et al., who used a list of 12 negative life events and assessed depression with BDI, followed by 

Honkalampi et al., who used a similar list of 12 negative life events and also BDI, 

followed by the two studies assessing depression with structured diagnostic interview instead of screening, which are 

Assari & Lankarani, who used a list 8 stressful life events and 

Maciejewski et al., who used a list of 11 negative life events. 

These studies were also inserted in the revised introduction section in the same order. 

Reviewer #1

1. The introduction opens with a reference to ongoing COVID pandemic as an example of an important, life-changing event (lines 47-48), which sets the study up as though it will measure the effect of the pandemic on depressive symptoms. However, it appears that the data for this study were in fact collected many years before the pandemic began. It would be good to make clearer in the introduction that this study is not about the pandemic specifically.

We deleted the reference to COVID since our study is not about the pandemic specifically. 

2. In general, the introduction does not describe the context for the study in enough detail. Information about previous studies that have examined the association between life events and depression is scarce (refs 5-9 on lines 59-62), giving the impression that there is relatively little existing research in this area. For example, the authors state that previous studies have not generally examined the effect of sex on this relationship (lines 78-80). However, in the discussion, two sex effects in previous studies are mentioned for the first time (lines 332-334 and lines 341-342), and several more details about these and other previous studies are presented. These details were needed in the introduction, to make it clear how the current study builds on previous work.

We now describe the context for our study in more detail in our revised introduction. 

3. In lines 78-81 and line 90, age is mentioned as a potential moderating factor and a main focus of this study. However, although age was included in the models, it was not discussed in any detail.

To the best of our knowledge, there is no previous evidence on the moderating effect of age on the association between life event stress and depression, but there are a few studies examining the moderating effect of age on the association between number of life events or presence of specific life events and depression, which we now included in the revised introduction and discussion. 

4. In the method section the authors state that the data for this study were drawn from a previous study for which participants were recruited around 20 years ago, with follow-ups at 5 and 10 years (lines 94-98). It is not clear when the data for this study were actually collected – at the outset of the original study, in a follow-up examination, or across multiple time points?

This study is based on a cross-sectional analysis of baseline data (collected between the years 2000 and 2003 at the outset of the study), which we now clarified in our revised method section. 

5. It is not entirely clear what instructions participants were given for the self-administered questionnaire (lines 104-108). Were the examples of important life events on line 106 provided to participants as examples, or were the participants left to decide on their own what constituted an important life event (this seems to be what is implied by “open response format” on line 108/113-114)? If left to decide on their own, is it possible that women/men and optimists/pessimists used different criteria to determine what constituted an important life event?

The original instructions were as follows (original in German): 

Translated into English, this would be: “Have there been any events in the past 6 months that were particularly important to you or that changed your life? (e.g. death or serious illness of a loved one, serious career change, separation, relocation)” Participants had to tick a box for “No” or “Yes” and if the ticked “Yes” they could freely write down these events on the lines. We now clarified the instructions in our revised method section. Due to the open response format, it is possible that women/men and optimists/pessimists used different criteria to determine what constituted an important life event. We added this to our limitations section. 

6. On lines 112-115 the authors argue that their open response format offers more flexibility than using a checklist of life events, yet participants’ responses were evaluated according to a checklist of events for which normative data on severity were available. It is not clear why the open response format is therefore more flexible. Perhaps the authors could expand on this idea. To what extent were participants left to self-determine what constituted an important life event?

As explained above, participants could freely write down everything they regarded as an important life event, which we now added to the method section. Not having a checklist with predefined life events is more efficient for the participants since they do not have to go through a long list with lots of events that do not apply and they do not become biased in any direction, which offers them more flexibility for their answers. Regarding the evaluation of participant responses, we chose the SRRS, which is widely used in life event research and offers the possibility of continuous assessment of stress caused by life events. Another possibility would be to ask how much stress the participants experienced by each of the life events they wrote down to get a subjective evaluation of life event stress, however, this might also be influenced by the current emotional state of the participant. 

7. Was there a reason why all of the life events were evaluated by only one rater (lines 108-110)? How can the authors be sure that the rater’s evaluations were reliable? Given the flexible open-response format, did any participants report life events that were not on the existing checklist of events? If so, how were these responses dealt with?

Due to the large sample size, it was not possible to have all participant responses rated by several raters. We had a small sample (n=20 participants who reported life events) rated by a second rater, which resulted in a high interrater reliability for the SRRS score (interrater correlation = 0.86). Less than 1% of all participant responses regarding concrete life events could not be scored with the SRRS, this was mostly because lack of information (e.g., just the word “father” or only a date was written down). These participants who gave insufficient information to score life events were included in the analysis of the presence of life events but were not included in the analysis of the SRRS score because they had a missing value here. Due to the very low number of missings, we do not believe that this will distort our observations to a relevant degree. 

8. Since the comparisons of interest in this study were women vs. men and optimists vs. pessimists, it would have been helpful to see some data on the SRRS scores and CES-D scores in these different subgroups. Were optimists and pessimists equally likely to report important life events, for example?

We added this data to the Supplement (Supplemental Table S1) as suggested. As already indicated in the “Interacting influence of the presence of important life events/life event stress with age, sex, and optimism on depressive symptoms” sections, women and pessimists report more important life events and show higher depression scores than men and optimists, however, next to this main effect, the significant interaction term reveals that the influence of life events on depression is stronger in women than in men and stronger in pessimists than in optimists. 

9. The argument is made that both positive and negative life events can be experienced as stressful due to the need to readjust (e.g., lines 72-73, lines 126-127). Following on from point 6 above, is it possible that, for example, optimists were more likely to report positive important life events, and this could explain the weaker relationship with depressive symptoms?

As shown in the new Supplemental Table S1, pessimists reported slightly more often important life events but had lower stress scores than optimists. When we look at single life events, pessimists report only slightly more often specific negative life events such as marital separation and fired at work. We now show presence of all important life events of the SRRS in the original order stratified by age, sex, and optimism in Supplemental Tables S2-4. 

10. In the abstract, the authors state that they examine the presence of important life events and the RESULTING life event stress (line 26). Similarly, elsewhere the argument is made that higher stress is associated with higher depression scores (e.g., line 310-312). However, since the study did not directly measure stress, it is not clear that the participants in this study experienced stress, only that these life events tend to be stressful. These claims could be worded a little more carefully, to reflect what was actually measured.

We now deleted the term “resulting” and rephrased the term “stress” as suggested. As explained in section 2.2.1, the SRRS does not measure individual stress responses but is suggested to measure life event stress because the adaptations required after the experience of a life changing event are thought to be stressful for most individuals. 

11. The authors’ stance on the causality of the relationship between life events and depression was unclear. In lines 52-55 of the introduction, they state that a robust and causal association between stressful life events and depression has become established, whereby stressful life events trigger depression. Later, it is acknowledged that the current study is unable to draw conclusions about causality, and that the authors rely heavily on other studies to conclude that stressful life events cause depression (line 376). The arguments in the conclusion then appear to depend on the causality evidenced by other studies, rather than on the data presented in the current study. Given that this study is apparently the first to measure the moderating effects of sex and optimism, how can we be confident that this assumption of causality is justified?

In our study, we cannot show causal relationships, only associations, since our study is a cross-sectional observational study. As explained in our revised introduction, previous studies with more complex study designs like longitudinal analyses of population-based samples including dizygotic and monozygotic twins have made it possible to conclude that life events actually trigger, that is causally influence depressive reactions instead of just being symptoms of depression. However, these studies did not include continuous scores of life event stress and did not analyze moderators of the associations between life events and depression. We now clarified that our study is not suited to draw causal conclusions in our revised limitations section. To prevent confusion, we deleted the section about reliance on previous studies. 

12. In the discussion section, there is a list (beginning on line 315) of several previous studies that have explored similar research questions, albeit using different scales, sampling methods, etc. Some of the findings of these previous studies were consistent with the current findings, while others were inconsistent. The list format is quite repetitive, and it is difficult to follow the argument throughout – what is the overall message? The argument would be clearer if the findings from the previous studies could be aggregated in a meaningful way, such as by the methods used, the time period sampled, etc.

We now ordered the previous population-based studies by the methods used beginning with:

Dalgard et al., who used a list of 12 negative life events and assessed depression with BDI, followed by 

Honkalampi., et al who used a similar list of 12 negative life events and also BDI, 

followed by the two studies assessing depression with structured diagnostic interview instead of screening, which are 

Assari & Lankarani who used a list 8 stressful life events and

Maciejewski et al who used a list of 11 negative life events. 

These studies were also inserted in the revised introduction section in the same order. 

13. It is not clear why a less comprehensive assessment of life events using a checklist of 8 events vs. free response would potentially reverse the sex effect (lines 334-335). This argument needs to be made more explicit. The same applies to the potential difference between clinical and subclinical depression (lines 336-338).

Men tend to report different life events than women, which we now also show in our new Supplemental Table S3 and as it is operationalized in the SRRS, different life events have different stress-inducing properties so that for example the experience of the death of a spouse has a high probability of inducing a clinical depression whereas minor violations of laws rather lead to short-term negative emotions. Moreover, specific life events are more harmful for one sex than another, i.e., job loss is usually more severe for men than for women while loss of a family member is usually more severe for women than for men. With short life event checklists, there is a higher probability of having an overrepresentation of life events more often experienced by one sex or overrepresentation of life events leading to clinical vs subclinical depression. We now added these arguments to our discussion section. 

14. On lines 356-363, there is a section on previous uses of SRRS (refs 11, 12, 13) showing that depressives had higher stress scores (i.e., more/more serious life events) than controls in 2/3 studies, and that stress score associated with depression severity in one study. This information would be better in the introduction (see point 2, above).

We now present this information in the revised introduction section as suggested. 

15. On lines 378-380, the authors argue that even if the causality is reversed, such that depression leads to the experience of stressful life events, “this still has important clinical implications in a way that the experience of life events must be considered more in the prevention and treatment of depression”. This argument was difficult to follow; if depression causes stressful life events, how could consideration of the experience of life events prevent or treat depression?

It is important because it is a vicious cycle: depressive patients experience more negative life events, which prevents remission and leads to recurrent depressive episodes. In depression treatment, it is important to convey this knowledge in psychoeducation, identify life events leading to recurrent depressive episodes and work on ways to interrupt this vicious cycle. We now added this information to our discussion section. 

16. A new study is introduced in the conclusion on lines 389-396. It would be better to include this argument in the main body of the discussion so that the conclusion remains focused on the current investigation.

We now included this argument in the new “clinical implications” section of our revised discussion. 

17. Minor points: 

Line 84-85 reads “From the clinical point of view, especially subclinical depression is important because it is associated with…”; it would read better as “From a clinical point of view, subclinical depression is especially important because….”

Changed as suggested. 

Lines 152-155 – this sentence is quite long, and would read better with commas.

Changed as suggested. 

Lines 310-311 – “In case important life events were experienced” would read better as “In participants who experienced important life events”

Changed as suggested. 

Line 311-312 – “higher stress… was again significantly associated…” – the “again” appears to be redundant here

Changed as suggested. 

Line 313 – “Women and individuals with pessimistic personality were more susceptible to the negative influence of life events on depression”. The wording is ambiguous, as it seems to suggest there is a negative association between life events and depression, rather than the positive association that was observed.

We changed “negative” into “adverse”.

Line 317 – “The so far largest population-based study” would read better as “So far, the largest population-based study”

Changed as suggested. 

Line 350 – “this study could show….” presumably means “this study showed that….”?

Changed as suggested.

---

## [Decision Letter · Decision Letter 1]

9 May 2022

PONE-D-21-02934R1Association between life events and later depression in the population-based Heinz Nixdorf Recall study – the role of sex and optimismPLOS ONE

Dear Dr. Gronewold

Thank you for submitting your manuscript to PLOS ONE. After careful consideration, we feel that it has merit but does not fully meet PLOS ONE’s publication criteria as it currently stands. Therefore, we invite you to submit a revised version of the manuscript that addresses the points raised during the review process. Both one reviewer and I have closely examined your revised manuscript. The main theoretical  point that needs be addressed is whether stronger association between life event stress and depression in the pessimist group is due to increased statistical power because the sample size is larger and therefore the impact this would have on your conclusions.  There  also needs to be more detail about the second rater coding and also the reliability calculations for the scoring of life events. A reference is needed  to support the argument that life events have more impact on men than women in line 392.   A few minor typographical/wording issues are  also outlined below in the comments of Reviewer 1. Please address all these points in your revision. 

 Please submit your revised manuscript by June 23, 2022. If you will need more time than this to complete your revisions, please reply to this message or contact the journal office at plosone@plos.org. Please include the following items when submitting your revised manuscript:A rebuttal letter that responds to each point raised by the academic editor and reviewer(s). You should upload this letter as a separate file labeled 'Response to Reviewers'.A marked-up copy of your manuscript that highlights changes made to the original version. You should upload this as a separate file labeled 'Revised Manuscript with Track Changes'.An unmarked version of your revised paper without tracked changes. You should upload this as a separate file labeled 'Manuscript'.If applicable, we recommend that you deposit your laboratory protocols in protocols.io to enhance the reproducibility of your results. Protocols.io assigns your protocol its own identifier (DOI) so that it can be cited independently in the future. For instructions see: https://journals.plos.org/plosone/s/submission-guidelines#loc-laboratory-protocols. Additionally, PLOS ONE offers an option for publishing peer-reviewed Lab Protocol articles, which describe protocols hosted on protocols.io. Read more information on sharing protocols at https://plos.org/protocols?utm_medium=editorial-email&utm_source=authorletters&utm_campaign=protocols.

We look forward to receiving your revised manuscript.

Kind regards,

Barbara Dritschel, PhD

Academic Editor

PLOS ONE

Journal Requirements:

Reviewers' comments:

Reviewer's Responses to Questions

**Comments to the Author**

1. If the authors have adequately addressed your comments raised in a previous round of review and you feel that this manuscript is now acceptable for publication, you may indicate that here to bypass the “Comments to the Author” section, enter your conflict of interest statement in the “Confidential to Editor” section, and submit your "Accept" recommendation.

Reviewer #1: (No Response)

2. Is the manuscript technically sound, and do the data support the conclusions?

Reviewer #1: Yes

3. Has the statistical analysis been performed appropriately and rigorously? 

Reviewer #1: Yes

4. Have the authors made all data underlying the findings in their manuscript fully available?

Reviewer #1: No

5. Is the manuscript presented in an intelligible fashion and written in standard English?

Reviewer #1: Yes

6. Review Comments to the Author

Reviewer #1: The authors have addressed the previous points that were raised, and the re-working of the introduction and discussion makes the message much clearer. The inclusion of the additional information in the supplementary tables is also very useful. I have one main question arising from the inclusion of these new tables:

- Table S4 shows the frequency of different life events stratified by optimism, and shows that the cut off for determining whether a participant is an optimist or a pessimist results in many more pessimists than optimists (i.e., 0.8% of the optimist group is 6 participants, but 0.8% of the pessimist group is 32 people). Is it possible that the stronger association between life event stress and depression in the pessimist group is due to increased statistical power because the sample size is larger? To put it another way, does the association look smaller in the optimist group because of lower power? If this is a possibility, how would it affect the conclusions?

Aside from that question, I have just a few minor comments after my second reading of the manuscript. In the following comments, line numbers refer to the line numbers in the revised manuscript.

1. line 70: "self-report screenings which are often not feasible in large population based studies" - I think the authors are arguing that self-report measures are usually used in larger studies because structured interviews are usually not feasible, but the wording makes it sound like it's the self-report measures that are not feasible.

2. lines 151-152: I think it would be good to include the details of the second-coding and reliability analysis for the life event scoring in here, so that readers can see that the scoring of life events was carried out more rigorously.

3. line 175: "optimism is regarded AS a stable trait" - the word as is missing

4. line 327: stronger influence of stress on optimists compared to pessimists - this appears to be the opposite of what is claimed elsewhere

5. line 392-393: an argument is made about some life events having more impact on men vs. women, and vice versa. A reference is needed here

7. PLOS authors have the option to publish the peer review history of their article (what does this mean?). If published, this will include your full peer review and any attached files.

Reviewer #1: No

---

## [Author Response · Author response to Decision Letter 1]

7 Jun 2022

Gronewold J, et al.

‘Association between life events and later depression in the population-based Heinz Nixdorf Recall study – the role of sex and optimism’ (MS ID: PONE-D-21-02934R1) 

Responses to reviewers

Please note that all new/revised text is highlighted in yellow in the manuscript.

Reviewer #1

The authors have addressed the previous points that were raised, and the re-working of the introduction and discussion makes the message much clearer. The inclusion of the additional information in the supplementary tables is also very useful. I have one main question arising from the inclusion of these new tables:

Table S4 shows the frequency of different life events stratified by optimism, and shows that the cut off for determining whether a participant is an optimist or a pessimist results in many more pessimists than optimists (i.e., 0.8% of the optimist group is 6 participants, but 0.8% of the pessimist group is 32 people). Is it possible that the stronger association between life event stress and depression in the pessimist group is due to increased statistical power because the sample size is larger? To put it another way, does the association look smaller in the optimist group because of lower power? If this is a possibility, how would it affect the conclusions?

In Table S4, the number and percentage of optimists and pessimists reporting different life events is shown. 6 participants of the optimist group (0.8%) reported that they experienced the life event “death of a spouse”, which was reported by the same percentage of participants from the pessimist group (0.8%, n=32). Since we defined optimists as subjects with Life Orientation Test-Revised score above the upper quartile (>Q3) and pessimists as subjects with Life Orientation Test-Revised score below the upper quartile (≤Q3), we have a smaller sample size in the optimist group (n=781) than in the pessimist group (n=3820). Even though we thus have higher statistical power in the pessimist group, resulting in narrower confidence intervals and lower p-values, the effect estimates for the association of life event stress and depression clearly differ between optimists and pessimist with the unstandardized regression weight from linear regression being 0.9 in optimists and 2.3 in pessimists (Table 3). According to up-to-date statistical recommendations, relevant effect modification is assumed in case of a difference in effect estimates of more than 10%. In our study, we have a difference of 55%, which clearly indicates that the effect of life events on depression is modified by the character trait optimism vs pessimism. This is also supported by the observed statistically highly significant interaction term (p<0.001) between life event stress and the character trait optimism vs pessimism (line 325). Consequently, the different sample size of optimists and pessimists does not affect our conclusion of a stronger association between life event stress and depression pessimists. 

Aside from that question, I have just a few minor comments after my second reading of the manuscript. In the following comments, line numbers refer to the line numbers in the revised manuscript.

1. line 70: "self-report screenings which are often not feasible in large population based studies" - I think the authors are arguing that self-report measures are usually used in larger studies because structured interviews are usually not feasible, but the wording makes it sound like it's the self-report measures that are not feasible.

The reviewer is completely right, that sentence was misleading, we changed it to “The minority of population-based studies assessed depression with structured diagnostic interviews because – compared with self-report screenings – structured diagnostic interviews require a lot more time and psychiatrically trained staff and thus are often not feasible in large population-based studies designed to address multiple research questions.

2. lines 151-152: I think it would be good to include the details of the second-coding and reliability analysis for the life event scoring in here, so that readers can see that the scoring of life events was carried out more rigorously.

We now included details of the second-coding and reliability analysis as suggested (line 152/153, line 172).

3. line 175: "optimism is regarded AS a stable trait" - the word as is missing

Changed as suggested. 

4. line 327: stronger influence of stress on optimists compared to pessimists - this appears to be the opposite of what is claimed elsewhere 

That was a mistake, we changed it to “This interaction revealed the stronger influence of life event stress on pessimists compared to optimists.”

5. line 392-393: an argument is made about some life events having more impact on men vs. women, and vice versa. A reference is needed here

Reference inserted as suggested.

---

## [Editor Report · Decision Letter 2]

7 Jul 2022

Association between life events and later depression in the population-based Heinz Nixdorf Recall study – the role of sex and optimism

PONE-D-21-02934R2

Dear Dr. Gronewold,

We’re pleased to inform you that your manuscript has been judged scientifically suitable for publication and will be formally accepted for publication once it meets all outstanding technical requirements.

Kind regards,

Barbara Dritschel, PhD

Academic Editor

PLOS ONE
---

## [Editor Report · Acceptance letter]

11 Jul 2022

PONE-D-21-02934R2 

Association between life events and later depression in the population-based Heinz Nixdorf Recall study – the role of sex and optimism 

Dear Dr. Gronewold:

I'm pleased to inform you that your manuscript has been deemed suitable for publication in PLOS ONE. Congratulations! Your manuscript is now with our production department. 

Kind regards, 

on behalf of

Dr. Barbara Dritschel 

Academic Editor

PLOS ONE